# Clinical *Burkholderia pseudomallei* isolates from north Queensland carry diverse $bimA_{Bm}$ genes that are associated with central nervous system disease and are phylogenomically distinct from other Australian strains

**Delaney Burnard**[1], **Michelle J. Bauer**[1], **Caitlin Falconer**[1], **Ian Gassiep**[1,2], **Robert E. Norton**[3], **David L. Paterson**[1], **Patrick NA Harris**[1,4]*

**1** University of Queensland Centre for Clinical Research, Royal Brisbane and Woman's Hospital, Herston, Queensland, Australia, **2** Department of Infectious Diseases, Mater Hospital Brisbane, South Brisbane, Queensland, Australia, **3** Pathology Queensland, Townsville University Hospital, Townsville, Queensland, Australia, **4** Pathology Queensland, Royal Brisbane & Women's Hospital, Herston, Queensland, Australia

* p.harris@uq.edu.au

## Abstract

### Background

*Burkholderia pseudomallei* is an environmental gram-negative bacterium that causes the disease melioidosis and is endemic in many countries of the Asia-Pacific region. In Australia, the mortality rate remains high at approximately 10%, despite curative antibiotic treatment being available. The bacterium is almost exclusively found in the endemic region, which spans the tropical Northern Territory and North Queensland, with clusters occasionally present in more temperate climates. Despite being endemic to North Queensland, these infections remain understudied compared to those of the Northern Territory.

### Methodology/Principal findings

This study aimed to assess the prevalence of central nervous system (CNS) disease associated variant $bimA_{Bm}$, identify circulating antimicrobial resistance mutations and genetically distinct strains from Queensland, via comparative genomics. From 76 clinical isolates, we identified the $bimA_{Bm}$ variant in 20 (26.3%) isolates and in 9 (45%) of the isolates with documented CNS infection (n = 18). Explorative analysis suggests a significant association between isolates carrying the $bimA_{Bm}$ variant and CNS disease (OR 2.8, 95% CI 1.3–6.0, P = 0.009) compared with isolates carrying the wildtype $bimA_{Bp}$. Furthermore, 50% of isolates were identified as novel multi-locus sequence types, while the $bimA_{Bm}$ variant was more commonly identified in isolates with novel sequence types, compared to those with previously described. Additionally, mutations associated with acquired antimicrobial resistance were only identified in 14.5% of all genomes.

**Data Availability Statement:** Genomes derived from this study are publicly available at NCBI BioProject ID: PRJNA717363.

**Funding:** This study has been funded by a 2020 Research Grant from the European Society of Clinical Microbiology and Infectious Diseases (ESCMID; https://www.escmid.org/) awarded to DB. The funder played no role in the study design, data collection, analysis or drafting of the manuscript.

**Competing interests:** I have read the journal's policy and the authors of this manuscript have the following competing interests: DLP has received research support from Shionogi, MSD and Pfizer and honoraria for speaking or advisory boards from MSD, Pfizer, Sumitomo, Biomerieux, Accelerate and Lyosvant. PNAH reports grants from Shionogi, Merck (MSD) and Sandoz and honoraria for speaking or advisory boards from Pfizer, Sandoz and Sumitomo. All other authors declare no conflicts of interest.

## Conclusions/Significance

The findings of this research have provided clinically relevant genomic data of *B. pseudomallei* in Queensland and suggest that the *bimA$_{Bm}$* variant may enable risk stratification for the development CNS complications and be a potential therapeutic target.

## Author summary

Melioidosis is a life-threatening infection, caused by the Gram-negative bacterium *Burkholderia pseudomallei*, which is endemic to tropical regions in Australia. Variants of the *bimA* gene have been proposed as a virulence factor associated with more severe disease. In a genomic analysis of 76 clinical *B. pseudomallei* isolates from Queensland, Australia, we identified that the *bimA$_{Bm}$* variant was associated with infection involving the central nervous system (odds ratio 2.8, 95% Confidence Interval: 1.3–6.0, P = 0.009), compared to isolates with the wild-type allele *bimA$_{Bp}$*. Half of the isolates from this region were novel multi-locus sequence types, and *bimA$_{Bm}$* was more commonly seen in these novel sequence types. Early genomic characterisation to identify virulence factors such as *bimA$_{Bm}$*, may be useful as an early marker of more complex disease that could guide further investigation and help determine optimal treatment. Further investigation of a "genomics-guided" approach to the clinical management of this complex infectious disease are warranted.

## Introduction

*Burkholderia pseudomallei* is an environmental gram-negative pathogen present in Asia, South America, Africa and the Pacific. The resulting disease, melioidosis, is endemic across northern tropical Australia, with occasional clusters appearing in more temperate climates such as Western Australia, Southeast Queensland and New South Wales. Studies of melioidosis in Australia are disproportionally inclusive of the Top End of the Northern Territory and its surrounds. This geographic area is made up of approximately one quarter indigenous Australians, most of which live in small remote communities/towns [1]. Indigenous Australians in this area are significantly more affected and have a greater burden of disease [2,3], while melioidosis has also been reported as the leading cause of fatal community-acquired bacteremic pneumonia in the region's largest hospital [1].

The disease melioidosis, caused by *B. pseudomallei* infection, can present with an extensive range of disease manifestations including, but not limited to, pneumonia, sepsis, and skin and soft tissue abscesses [4]. Successful treatment can be compromised by the presence of CNS disease (e.g. encephalomyelitis, abscess, meningitis, cranial nerve impairment) [5]. As these infections are often sub-clinical, exhibit varying times to clinical presentation, have diverse clinical presentations and affect multiple other organs; they can be challenging cases to manage [5,6].

CNS infections have been associated with the virulence factor *bimA$_{Bm}$*, a variant of the wild-type, *bimA$_{Bp}$* [7–9]. These *bimA* genes encode as intracellular motility factor A, type V effector proteins that utilise host cellular actin. This motility results in the ability to evade the host immune system and invade the CNS via inter-cellular host cell migration [10]. The *bimA$_{Bm}$* variant is significantly truncated to that of *bimA$_{Bp}$* and is significantly more similar to *bimA$_{Ma}$*, of the highly infectious *B. mallei*, that is fatal in humans if left untreated [11]. Furthermore, the carboxy-terminals of the *bimA* genes of both species are also similar to the YadA

auto-secreted adhesion protein of *Yersinia enterocolitica*, another highly pathogenic bacterial species [10,12,13]. Patients with infections possessing the $bimA_{Bm}$ gene are suggested to be 14 times more likely to present with CNS disease, while the wild-type variant $bimA_{Bp}$ is linked to pneumonia [7,14].

The $bimA_{Bp}$ gene has been identified in both clinical and environmental isolates from the Northern Territory; yet isolates from North Queensland have not been as extensively characterised, despite both states being endemic regions [7,14] and tropical far north Queensland showing similar disproportionate patterns of infection and disease burden in indigenous Australian populations, to those reported in the Northern Terrritory [15,16]. Clinical presentations of CNS disease have been documented in a retrospective Queensland isolate collection [17]; however, the diversity of $bimA_{Bm}$ and the association with central nervous system disease in Queensland remains unknown.

This study aimed to use comparative genomics to screen for the presence of the virulence factor $bimA_{Bm,}$ which has been previously linked to CNS disease in *B. pseudomallei* infections in patients of the Northern Territory. Furthermore, genomic diversity and antimicrobial resistance profiles are described.

## Methods

### Ethics statement

This study was performed under ethics approval from the Human Research and Ethics Committee of the Royal Brisbane and Women's Hospital, as low or negligible-risk with waiver of patient consent (LNR/2020/QRBW/65573); site specific authority was obtained from the Townsville Hospital and Health Service with approval under the Queensland Public Health Act to access de-identified patient data.

## Isolates

Clinical *B. pseudomallei* isolates have been collected prospectively over the last 22 years (1996–2018) from participating hospital pathology facilities (Cairns, Townsville and Central laboratories of Pathology Queensland). A total of 76 isolates were selected between postcodes 4895–4500 to include isolates from all of Queensland, from a collection of 400 clinical isolates. All isolates from patients with documented CNS involvement were included, which consisted of, six blood, four skin and soft tissue, four cerebral spinal fluid, two brain, and two sputum isolates(n = 18). The remaining 58 isolates were selected to include a variety of other common *B. pseudomallei* clinical presentations (Table 1). An additional 24 publicly available reference genomes were included in this study, of these, 13 were derived from the Northern Territory, eight from Queensland and one representative each from Papua New Guinea, China and Thailand (Table C in S1 Text).

## DNA extraction and WGS

The 76 isolates were recovered from -80˚C storage and subcultured twice to ensure purity. DNA extraction was performed with the QIAGEN DNAeasy ultra-pure DNA extraction kit according to manufacturer's instructions. Sequencing libraries were generated using the Nextera Flex DNA library preparation kit and sequenced on the MiniSeq System (Illumina Inc., San Diego, CA, USA) on a high output 300 cycle cartridge according to the manufacturer's instructions. Five strains with unique $BimA_{Bm}$ sequence and/or genomic diversity were prepped for long read sequencing using the MinION (Oxford Nanopore Technologies, Oxford, UK), where sequencing libraries were generated using the Rapid Barcoding

**Table 1. Metadata regarding the *B. pseudomallei* isolates included in this study.**

| *B. pseudomallei* isolates | |
|---|---|
| Dates collected | 1996–2018 |
| Geographical range | 4000–4895 Queensland, Australia |
| gender | 31 F, 45 M |
| Age range | 6–84 years |
| Central Nervous System Disease | 18 (23.7%) |
| Mortality* | 15 (19.7%) |
| Isolation sites: | |
| Blood | 42 |
| Pus | 8 |
| Sputum | 7 |
| Cerebral spinal fluid | 4 |
| Tissue | 4 |
| Brain | 2 |
| Endotracheal aspirate | 2 |
| Bronchoalveolar Lavage | 1 |
| Gastrointestinal Tract | 1 |
| Liver | 1 |
| Lung aspirate | 1 |
| Lymph | 1 |
| Urine | 1 |
| Unknown | 1 |
| Total: | 76 |

*only information for 23/76 isolates was available.

Sequencing Kit (SQK-RBK004) and run on a flow cell R9.4.1 for 72 hrs. Data generated from this study is available under the NCBI accession PRJNA717363.

## Genomic analysis

Illumina reads were trimmed with Trimmomatic v0.36 [18] and quality assessed with multiQC [19], genomes were assembled with SPAdes v3.14.0 [20] and annotated with Prokka v1.13 [21]. Long reads were filtered, quality checked, assembled and polished with Illumina reads where applicable using the MicroPIPE pipeline [22].

All read mapping was performed with BWA-MEM [23]. Sequence types (STs) were determined with multi-locus sequence typing (MLST) [24] (https://github.com/tseemann/mlst) and reads mapped to alleles retrieved from pubMLST, where SNPs were suspected [24]. Genotypic antimicrobial resistance was determined with ArDaP [25]. Whole genome alignment (4,950,632bp) and phylogenomic analysis of the 24 reference genomes and 76 genomes derived from this study (*n* = 100) were achieved using parSNP [26] JModelTest[27] (TVM+F+I+G4) and IQtree [28] (1000 bootstrap (BS) replicates).

## Statistical analysis

Using Stata and the csi command for unstratified cumulative incidence data [29], and isolates with *bimA$_{Bm}$* or CNS presentation (*n* = 29), Odds Ratio and Chi-Square were calculated, with a P value <0.05 considered significant.

## Results

In total, we identified the *bimA_Bm* virulence factor in 20/76 Queensland isolates (26.3%), geographically limited to the northern half of Queensland (Fig 1). Of the 18 isolates with known CNS disease, nine were found to carry the *bimA_Bm* virulence factor and nine carried the wild type *bimA_Bp*. A total of 20 isolates were confirmed to carry the *bimA_Bm* virulence factor, of which eleven had no clinical record of CNS disease. Risk analysis supported *bimA_Bm* being associated with CNS infection (Odds Ratio 2.8, 95% CI 1.3–6.1, P = 0.009).

The variation observed in the 20 *bimA_Bm* protein sequences was significant amongst isolates, with 18 proteins possessing unique sequences, while none of the sequences produced in this study were identical to that of the reference strain MSHR668. The variation was observed almost only in the proline-rich region of the protein (90-160aa) (Fig 2). However, five isolates carried an amino acid variant ΔN213S (Fig 2A, TSV1, TSV152, TSV141, TSV164, CAM60), of which four isolates had documented CNS disease (one individual died before diagnosis, CAM60), and four cases resulted in death (one individual made an unexpected recovery while in intensive care, TSV 152). These isolates were collected from as early as 1996, with the latest in 2012 and were significantly geographically dispersed (350–850 km apart). Additionally, four

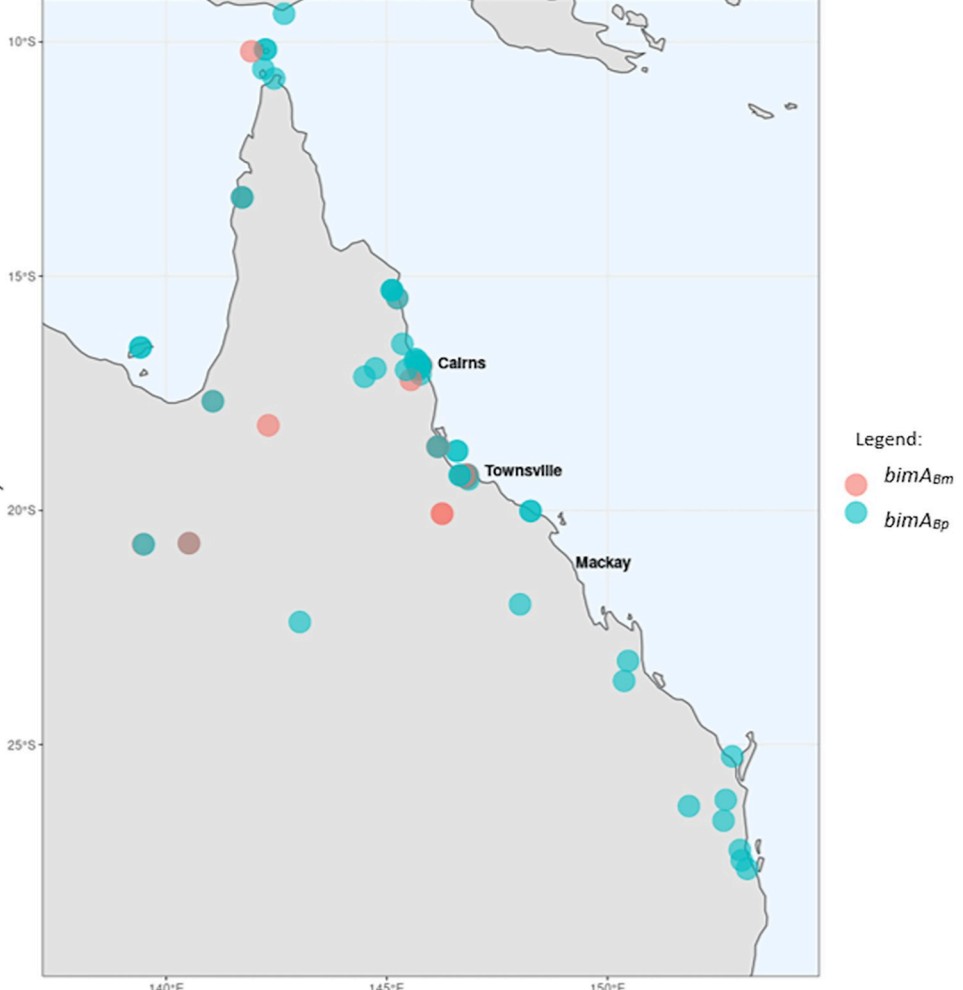

**Fig 1. Distribution of *B. pseudomallei* isolates with the *bimA_Bm* virulence factor in north Queensland (Generated using the rnaturalearth package in R: https://cran.r-project.org/web/packages/rnaturalearth/README.html).**

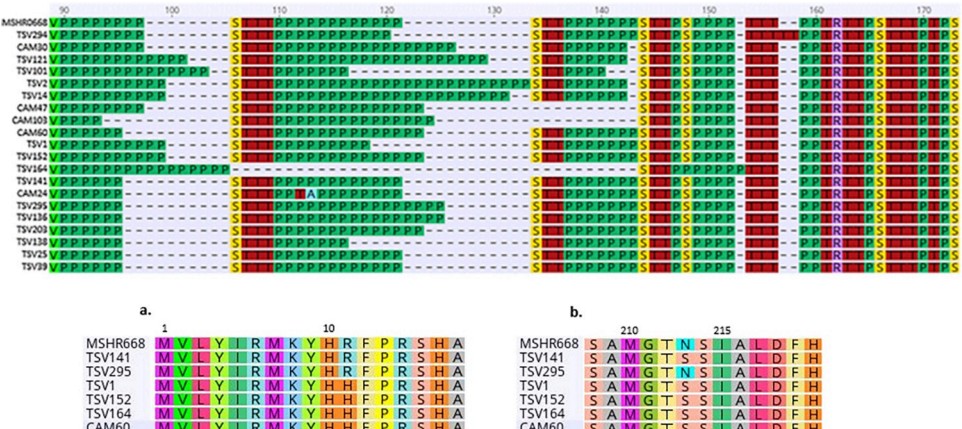

**Fig 2. Alignment of the *BimA<sub>Bm</sub>* proline-rich region identified in Queensland *B. pseudomallei* isolates.** Extensive variation of the proline rich region coloured green, is observed between isolates included in this study as well as compared to the Queensland *BimA<sub>Bm</sub>* reference genome MSHR668 located at the top of the alignment. **Fig 2A.** Alignment depicting the four isolates carrying an amino acid variant ΔR11H upstream of the proline rich region of the *BimA<sub>Bm</sub>* gene. **Fig 2B.** Alignment depicting the five isolates carrying an amino acid variant ΔN213S downstream of the proline rich region of the *BimA<sub>Bm</sub>* gene.

of these same isolates (TSV1, TSV152, TSV164, CAM60) and isolate TSV294 also possessed an upstream mutation ΔR11H (Fig 2B). Additionally, all isolates carrying *bimA<sub>Bm</sub>* also possess a truncated *bimC* gene which plays a role in intracellular spread and lies upstream of *bimA<sub>Bm</sub>* (S1 Fig).

From the 76 isolates we were able to identify 27 known multi-locus sequence types (STs) from 38 isolates (50%) (Table A in S1 Text). From these, at least three sequence types (109, 151, 1667) have also been identified in the Northern Territory. Of the 38 novel sequence types, 26 isolates were comprised of previously described alleles in a novel combination, and 12 were due to a SNP present in one of the seven alleles (Fig 3 and Table B in S1 Text). Novel allele variants were identified across six of the alleles, with the allele *ace* the only one without any variants (n = 12). In each novel case, sequence types were confirmed via read mapping to all seven alleles. In total 34 of the 38 isolates carried unique novel STs, as two of the STs occurred twice and one ST occurred three times in the sample set. Among the 20 isolates carrying the *bimA<sub>Bm</sub>* variant, 13 (65%) carried novel STs, over previously described STs (n = 7, 35%; Fig 3).

Phylogenomic analysis revealed the majority of Queensland isolates with novel STs branched significantly closer to the Queensland *bimA<sub>Bm</sub>* reference genome MSHR668, than to the Northern Territory and Asian genomes. This was also true for the *bimA<sub>Bm</sub>* variant. Two separate lineages can be identified in the phylogeny (point of divergence is approximately in the middle of the phylogeny), with the three clades at the bottom half of the phylogeny carrying 33 of the 38 novel STs identified (86.8%) and 18 of the 20 *bimA<sub>Bm</sub>* variants present in the sample set (90%). MLST diversity did not appear to be influenced by or correlated with the date the isolate was collected (Fig 3). No Queensland isolate was identical to that of a Northern Territory isolate, with limited clustering with Northern Territory or other Australian isolates. Additionally, five of the Queensland isolates included in this study clustered within the Australian/Asian clade. In this clade the reference genome K96243 from Thailand is the most basal, with BPC006 from China branching within the Australian isolates CAM2, 112, 189, TSV38 and 95. All of the isolates within this clade were of known ST and did not carry the *bimA<sub>Bm</sub>* variant.

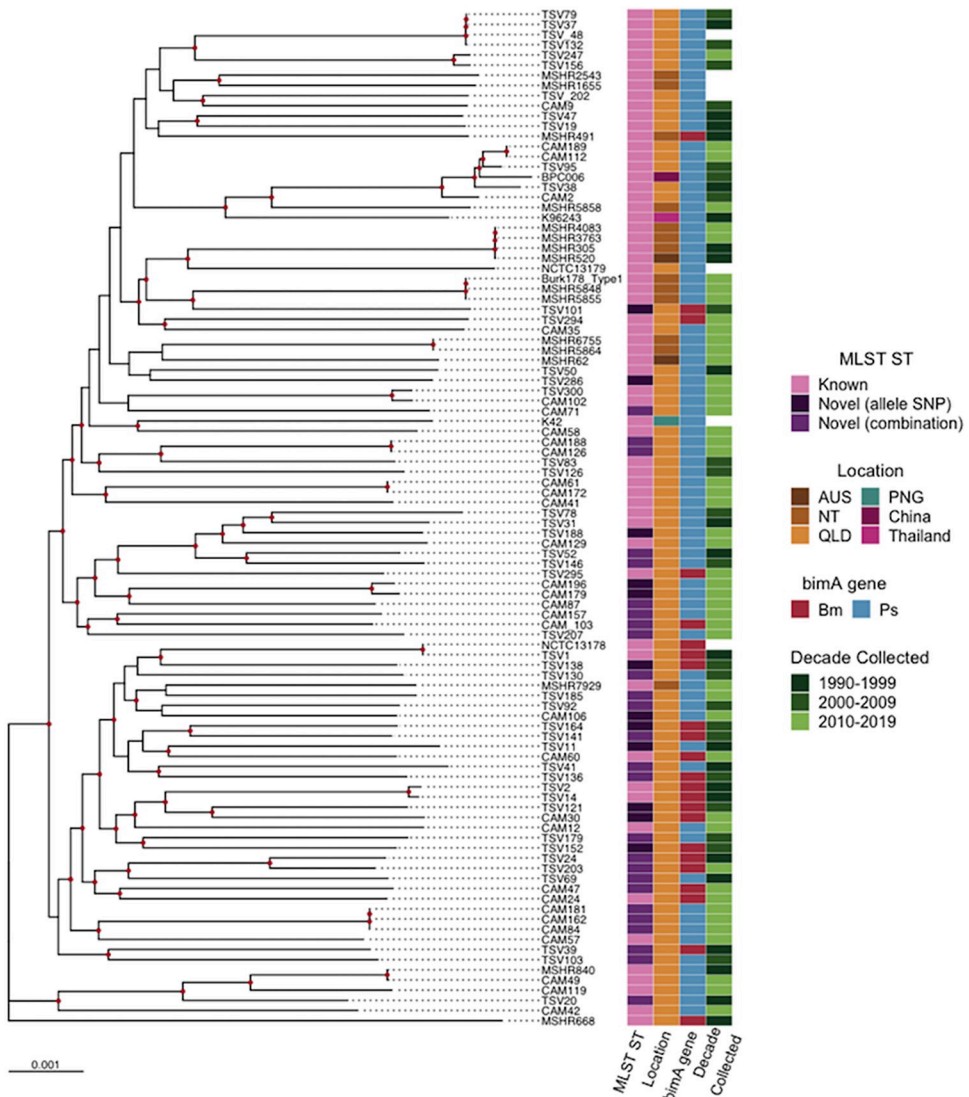

**Fig 3. Maximum likelihood phylogenomic tree of Australian *B. pseudomallei* isolates, rooted to reference MSHR668.** Bootstrap support ≥80 is shown at nodes in red and reference genomes are labelled with strain name stated in NCBI. Tree was built using IQtree 100 genomes and a 4,950,632 bp alignment generated from parSNP.

Approximately one third of the genomes did not contain any AMR related mutations at all (35.5%). The pre-cursor mutation to imipenem resistance (*penA* ΔT153A) was identified in 57.9% of all isolates. Only mutations associated with ceftazidime and meropenem resistance were predicted within the sample set. Ceftazidime resistance (loss of the PBP3 homolog BPSS1219) was only predicted from one isolate (1.3%), while meropenem resistance was predicted to be caused by the loss of function of the *amrR* efflux pump (BPSL1805) in 11 of the 76 isolates (14.5%). Meropenem resistance was encoded in 15% of $bimA_{Bm}$ isolates and 16% of $bimA_{Bp}$ isolates, respectively (Table 2).

## Discussion

Variation in the $bimA_{Bm}$ gene was present and extensive. Of the 76 isolates, 20 carried the virulence factor $bimA_{Bm}$ and only two of the sequences produced from this study were identical

**Table 2. Comparison of clinical presentation, sequence type and antimicrobial resistance within *bimA$_{Bm}$* or *bimA$_{Bp}$*-carrying *B. pseudomallei* isolates.**

| Variable | *bimA$_{Bm}$* Isolates (*n* = 20, (%)) | *bimA$_{Bp}$* Isolates (*n* = 56, (%)) |
|---|---|---|
| CNS presentation | 45 | 16 |
| Novel ST | 40 | 53.6 |
| Known ST | 60 | 46.4 |
| *amrR* mutation | 15 | 16 |
| Pre-curser mutation | 45 | 64.3 |
| No AMR mutation | 35 | 35.7 |
| Mortality* | 40 | 12.5 |

*information was only available for 23/76 isolates

(TSV39, TSV25). No sequence generated from this study was identical to any of the previously described reference sequences, suggesting depth of variation in the *bimA$_{Bm}$* gene is extensive and yet to be seen (Fig 2). Since the 2000's sequence variation of *BimA$_{Bm}$* in *B. pseudomallei* has not been discussed in great detail, despite being studied recently [8]. For example, five isolates possessed a ΔN213S substitution, positioned between the proline rich region and the YadA-like head domain, a highly conserved region [9]. Interestingly, the isolates with this mutation had a mortality rate of 80% and documented CNS disease at a rate of 80% (one patient died before CNS disease could be diagnosed). Unfortunately, similar to previous studies the effect these variants have upon the function of the protein and resulting virulence remain unclear [9]. Reference sequences from MSHR33, 491 and 172 all possess the mutation as well, suggesting this is not a recent event [9]. However, such high mortality rates and disease severity suggests that isolates with this type of mutation may be more virulent than those without.

Previous studies have suggested that *bimA$_{Bm}$* carrying isolates are 14 times more likely to develop CNS disease than the wild-type *bimA$_{Bp}$* isolates [7]. Explorative analysis of this data suggested the *bimA$_{Bm}$* variant and the development of CNS disease to have an Odds Ratio of 2.8 (95% CI 1.3–6.1 P = 0.009), compared to the wild type. However, as this dataset was selected for particular factors (CNS disease and postcode) these numbers should be viewed with caution. Perhaps what is more representative of the Queensland population is the 11 *bimA$_{Bm}$* variants identified from the 58 non-CNS disease isolates, suggesting 19% of *B. pseudomallei* infections will carry the virulence factor. Indeed, larger sample sizes collected at random will be needed to confirm these numbers and this should be considered for future studies. Sarovich *et al.*, also assessed the likelihood of clinical co-variants that may be associated with CNS disease, but did not find other associations [7]. Due to the lack of complete clinically relevant data surrounding the isolates in this study, we were unable to calculate if this was applicable to our results.

In this study, CNS disease isolates were evenly split between *bimA$_{Bp}$* and *bimA$_{Bm}$* genes and more isolates presented without CNS disease and *BimA$_{Bm}$* (n = 11), than with both of these variables (n = 9). Therefore, it is likely this gene is not the only factor driving the development of CNS disease. This was also evident in the Northern Territory, with 85.5% of isolates carrying the variant, not diagnosed with CNS disease. Additionally, 1% presented with CNS disease, but no variant [7]. In both studies this is a significant proportion of *bimA$_{Bm}$* isolates that have not developed CNS disease. An environmental study [14] of both *bimA$_{Bm}$* and another virulence factor lipopolysaccharide (LPS) suggested the *bimA$_{Bm}$* virulence factor was more likely to occur with LPS genotype B, the more prevalent genotype in Australia. Genotype A, is more

prevalent in Thailand and Southeast Asia, where CNS disease is less frequently reported [7,14]. Furthermore, the gene upstream of *bimA*, *bimC* was also assessed in this study. As both *bimA* and *bimC* genes have been shown to play a role in the intracellular spread of *B. pseudomallei* [30]. As all *bimA_Bm* isolates in this study possessed a truncated *bimC* gene, the affect this truncation has upon virulence and the role of *bimC* in intracellular spread remains unknown. However, it suggests the variant is significant to virulence in these strains and should be further investigated. There is no way to show a definitive correlation between these two virulence factors and *bimA_Bm* in Queensland without including both clinical and environmental isolates in future studies. The authors suggest a pan-genomic analysis coupled with extensive clinical metadata from as many isolates as possible representing both states evenly, to be the best approach at identifying all genes associated with CNS disease. It is possible that there may be some combination of genes derived only from the unique genetic diversity of Australian isolates responsible for the increased incidences of CNS disease observed here in Australia compared to those overseas.

Although Queensland has been included in MLST diversity analyses before, novel STs have not been described, which is significant given 50% of isolates in this study were novel STs (Fig 3) [31,32]. A large number of isolates (34%) were comprised of a previously undescribed combination of alleles to generate a novel ST (Table B in S1 Text). It also appears that isolates with the *bimA_Bm* virulence factor are more likely to be novel STs. This implies that Queensland isolates are exchanging genetic material [32]. The exchange of genetic material was also evidenced by the lack of clonality seen in both previously described and novel STs in Queensland. ST 70 was the most commonly described (6.5%) and was identified in both Brisbane and Cairns isolates suggesting one ST does not dominate certain spatial areas in this study, despite this being reported for the Northern Territory [14,31]. Furthermore, the same studies reported that Queensland and the Northern Territory do not share any common STs. However, in this study we identified at least three STs shared between the states (STs 109, 151, 1667), with the potential of more, as the exact location was not recorded for all STs in pubMLST. The identification of shared STs here, may be due to the previous under-representation, or exclusion of Queensland in *B. pseudomallei* isolates in previous studies [31–34].

Genomic analyses also identified the loss of function of the *amrR* regulator (amrAB-OprA efflux pump), associated with meropenem resistance in 14.5% of isolates, however this has not been validated phenotypically [25]. A pre-curser mutation in *penA* ΔT153A, was common amongst the isolates and in combination with other missense and promoter mutations can cause imipenem and amoxicillin-clavulanic acid resistance [25]. This is not overly worrisome as imipenem is not used to treat *B. pseudomallei* in Australia, meropenem is the preferred carbapenem, however amoxicillin-clavulanic acid is sometimes used in eradication therapy [6]. Phenotypically, antimicrobial resistance circulating in the Northern Territory comprises of approximately 3% for doxycycline and 0.9% for trimethoprim-sulfamethoxazole [35], while isolates remain susceptible to meropenem and ceftazidime [4,35,36]. This data suggests AMR may be rarely encountered in Queensland isolates and the current selection of antimicrobials for use against *B. pseudomallei* infections will be effective.

## Conclusion

This study has revealed a significant amount of genetic diversity in Queensland *B. pseudomallei* isolates, such as novel MLST sequence types and unique *bimA_Bm* gene sequences. The virulence factor *bimA_Bm* is linked to CNS disease, yet it is suspected there are multiple drivers for this type of infection. Further exploration into the virulence factors responsible for CNS disease should focus on maximum sample size, pan-genomics, detailed clinical metadata and

include environmental samples. Identification of CNS disease drivers may act as a screening test to warn clinicians and prompt additional investigations such as a lumbar puncture or MRI or ultimately provide a novel therapeutic target.

## Supporting information

**S1 Text. Supplementary tables. Table A in S1 Text.** Known sequence multi-locus types identified in Queensland *B. pseudomallei* isolates. **Table B in S1 Text.** Novel Queensland *B. pseudomallei* multi-locus sequence types, ~ flags the closest known allele. **Table C in S1 Text.** Details of the *Burkholderia pseudomallei* reference data used in this study.
(DOCX)

**S1 Fig. Read mapping of *bimC* demonstrating truncation in isolate TSV2 compared to *bimC* of reference genome BPSS1491.**
(PNG)

## Acknowledgments

The authors would like to thank Valentine Murigneux for the use of, and their assistance in their long-read assembly pipeline MicroPIPE and Dr Brian Forde for their assistance in constructing Fig 1.

## Author Contributions

**Conceptualization:** Delaney Burnard, Ian Gassiep, Robert E. Norton, David L. Paterson, Patrick NA Harris.

**Data curation:** Delaney Burnard, Ian Gassiep, Robert E. Norton.

**Formal analysis:** Delaney Burnard, Caitlin Falconer.

**Funding acquisition:** Delaney Burnard.

**Investigation:** Delaney Burnard, Michelle J. Bauer.

**Methodology:** Delaney Burnard, Michelle J. Bauer, Caitlin Falconer, Ian Gassiep.

**Project administration:** Delaney Burnard, Patrick NA Harris.

**Resources:** Patrick NA Harris.

**Supervision:** Robert E. Norton, David L. Paterson, Patrick NA Harris.

**Visualization:** Caitlin Falconer.

**Writing – original draft:** Delaney Burnard.

**Writing – review & editing:** Michelle J. Bauer, Caitlin Falconer, Ian Gassiep, Robert E. Norton, David L. Paterson, Patrick NA Harris.

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
