## [Decision Letter · Decision Letter 0]

15 Nov 2021

Dear Dr. Harris,

Thank you very much for submitting your manuscript "Clinical Burkholderia pseudomallei isolates from Queensland carry diverse bimABm genes that are associated with central nervous system disease and are phylogenomically distinct from other Australian strains" for consideration at PLOS Neglected Tropical Diseases. As with all papers reviewed by the journal, your manuscript was reviewed by members of the editorial board and by several independent reviewers. In light of the reviews (below this email), we would like to invite the resubmission of a significantly-revised version that takes into account the reviewers' comments. 

We cannot make any decision about publication until we have seen the revised manuscript and your response to the reviewers' comments. Your revised manuscript is also likely to be sent to reviewers for further evaluation.

Sincerely,

Adly M.M. Abd-Alla, Prof asso.

Associate Editor

Abdallah Samy

Deputy Editor

Reviewer's Responses to Questions

**Key Review Criteria Required for Acceptance?**

**Methods**

-Are the objectives of the study clearly articulated with a clear testable hypothesis stated?

-Is the study design appropriate to address the stated objectives?

-Is the population clearly described and appropriate for the hypothesis being tested?

-Is the sample size sufficient to ensure adequate power to address the hypothesis being tested?

-Were correct statistical analysis used to support conclusions?

-Are there concerns about ethical or regulatory requirements being met?

Reviewer #1: METHODS: How did the investigators select the control isolates (e.g., Queensland NON-CNS isolates)? Couldn't some apparent differences be introduced between the two groups simply by differences in era of enrollment, location within the geographically large state of Queensland, or even characteristics of the patient (immigrant status, race, gender, age, immunocompromise/diabetic state)?

Reviewer #2: Materials and Methods section for this study is straight forward. No issues.

**Results**

-Does the analysis presented match the analysis plan?

-Are the results clearly and completely presented?

-Are the figures (Tables, Images) of sufficient quality for clarity?

Reviewer #1: Results are clear and clearly presented.

Reviewer #2: #1) Lines 88-90 and Table 1. Which of the 18 isolates make up the CNS strains? 4 of the strains from CSF and 2 from the brain, but it was not clear where the other 12 come from as listed in Table 1. Also, on line 89, it lists 18 CSF strains and the remaining 57 but that adds up to 75. Line 86 refers to a total of 76 isolates.

#2) Figure 2. This reviewer would expand the legend description. It is not clear what I should be focused upon. Do the various colors represent something notable?

#3) Line 132. Points out the important amino acid variant N213S for the 5 strains. Should this be included/shown in Fig 2? 

#4) Lines 136-139. These results are discussed as being important but not shown.

**Conclusions**

-Are the conclusions supported by the data presented?

-Are the limitations of analysis clearly described?

-Do the authors discuss how these data can be helpful to advance our understanding of the topic under study?

-Is public health relevance addressed?

Reviewer #1: 259-262 significant amount of genetic diversity in Queensland B.

265 unclear if absence of bimABm could identify low risk of CNS involvement given only half of the patients with CNS infection had the variant

Reviewer #2: No comments for conclusions.

**Editorial and Data Presentation Modifications?**

Reviewer #1: (No Response)

Reviewer #2: #1. Line 37. Should multi-locus sequence types be abbreviated as MLST versus ST? In other places (line 259), the abbreviation is shown as MLST.

#2. Table 2. Title refers to bimAps. Should this be bimABp?

#3. Line 176. Should this be worded: “Twenty isolated carried the “altered” virulence factor”?

**Summary and General Comments**

Reviewer #1: This study aimed to identify prevalence of bimABm, which is associated with meningitis/CNS infection, with comparative genomics , 

has been previously linked to neurological disease in B. pseudomallei infections in patients of the Northern Territory

From the 76 isolates analyzed,the bimABm variant was identified in 20 (26.3%) isolates and 9 (45%) of the isolates with documented central nervous system infection (n=18).

A significant association was noted between isolates carrying the bimABm variant and central nervous system disease (OR 2.8, 95% CI 1.3-6.0, P=0.009) compared with isolates carrying the wildtype bimABp. 

This association, however, has been described for over a decade (cit. 7 Stevens Mol Microbiol 2005)

METHODS: How did the investigators select the control isolates (e.g., Queensland NON-CNS isolates)? Couldn't some apparent differences be introduced between the two groups simply by differences in era of enrollment, location within the geographically large state of Queensland, or even characteristics of the patient (immigrant status, race, gender, age, immunocompromise/diabetic state)?

120-126 Half of CNS isolates had bimABm; implying that it is not fully causative. 

More interesting is what homologous genes were present in the 9 isolates from CNS diseas who did NOT have the bimABm variant.

Further, why did the authors not perform in vitro phenotypic testing to establish phenotypic variation in host actin-mediated mobility between the identified variants?

Alternately they could perform in vivo RNA seq on the bacteria to evaluate potential upstream regulatory differences among the bim ABm variants NOT associated with meningitis; and also among the 9 meningitis-derived isolates that did not manifest the BimABm variant.

151-154 Unsurprising result given proximity of the regions. More interesting are the Thailand derived isolates. Could these have been described amongst recent immigrants

167-172 Why not phenotypic confirmation of the AMR findings? 

259-262 significant amount of genetic diversity in Queensland B.

265 unclear if absence of bimABm could identify low risk of CNS involvement given only half of the patients with CNS infection had the variant

Reviewer #2: The results are interesting but the manuscript could be tightened in a few areas to better relay the story. Details below.

#1) This reviewer feels more details, distinction, and appropriate references should be provided for bimAbm and bimABp. 

a. The description is limited to just one sentence lines 63-67. This section lacks a description of what the difference are in general between these two versions of bimA. 

b. Also, it may be a semantics issue, but is it appropriate to call this a mutant? Perhaps more appropriate would be variant or referring to those Bp strains with bimABm as a subset of a population. 

c. The sentence starting at line 63 should have an appropriate reference for it.

#2) As highlighted above, many of the relevant mutations discussed in the results were not shown in the figures or even listed as "data not shown".

PLOS authors have the option to publish the peer review history of their article (what does this mean?). If published, this will include your full peer review and any attached files.

Reviewer #1: No

Reviewer #2: No
---

## [Decision Letter · Decision Letter 1]

7 Mar 2022

Dear Dr. Harris,

Thank you very much for submitting your manuscript "Clinical Burkholderia pseudomallei isolates from north Queensland carry diverse bimABm genes that are associated with central nervous system disease and are phylogenomically distinct from other Australian strains" for consideration at PLOS Neglected Tropical Diseases. As with all papers reviewed by the journal, your manuscript was reviewed by members of the editorial board and by several independent reviewers. The reviewers appreciated the attention to an important topic. Based on the reviews, we are likely to accept this manuscript for publication, providing that you modify the manuscript according to the review recommendations. 

Sincerely,

Adly M.M. Abd-Alla, Prof asso.

Associate Editor

Abdallah Samy

Deputy Editor

Reviewer's Responses to Questions

**Key Review Criteria Required for Acceptance?**

**Methods**

-Are the objectives of the study clearly articulated with a clear testable hypothesis stated?

-Is the study design appropriate to address the stated objectives?

-Is the population clearly described and appropriate for the hypothesis being tested?

-Is the sample size sufficient to ensure adequate power to address the hypothesis being tested?

-Were correct statistical analysis used to support conclusions?

-Are there concerns about ethical or regulatory requirements being met?

Reviewer #2: (No Response)

Reviewer #3: 1. The overall study design is appropriate and adequately address the hypothesis.

2. My only concern is if the authors have conducted a simple antibiotic susceptibility test on the isolates, particularly those 76 clinical isolates. This simple experiment can perhaps give a clearer correlation to AMR related mutations analysis conducted by the authors.

3. Line 98 - Papua New Guinea

Reviewer #4: -Are the objectives of the study clearly articulated with a clear testable hypothesis stated?

Yes

-Is the study design appropriate to address the stated objectives? Yes

-Is the population clearly described and appropriate for the hypothesis being tested? Yes even if some missing data about the strains could have been vey important

-Is the sample size sufficient to ensure adequate power to address the hypothesis being tested? Yes 

-Were correct statistical analysis used to support conclusions? Yes

-Are there concerns about ethical or regulatory requirements being met? Not applicable

**Results**

-Does the analysis presented match the analysis plan?

-Are the results clearly and completely presented?

-Are the figures (Tables, Images) of sufficient quality for clarity?

Reviewer #2: (No Response)

Reviewer #3: 1. The results are clear and well presented.

2. Can the authors prove the correlation between the presence of AMR related mutations with the isolates phenotype (antibiotic susceptibility)?

Reviewer #4: -Does the analysis presented match the analysis plan? Yes

-Are the results clearly and completely presented? Yes

-Are the figures (Tables, Images) of sufficient quality for clarity? Yes

**Conclusions**

-Are the conclusions supported by the data presented?

-Are the limitations of analysis clearly described?

-Do the authors discuss how these data can be helpful to advance our understanding of the topic under study?

-Is public health relevance addressed?

Reviewer #2: (No Response)

Reviewer #3: The conclusions is well presented.

Reviewer #4: -Are the conclusions supported by the data presented? Yes

-Are the limitations of analysis clearly described? Could be improved

-Do the authors discuss how these data can be helpful to advance our understanding of the topic under study? Yes

-Is public health relevance addressed? Yes

**Editorial and Data Presentation Modifications?**

Reviewer #2: Several minor editorial suggestions for the authors.

Lines 35 & 59. Add abbreviate for CNS as needed.

Line 51. ....Territory and surrounds. Is something wrong/ missing with sentence?

Line 51. This geographic.... Should "area" be added?

Line 57. Something missing from the start of this sentence? "Infections cause the disease meliodosis....

Line 68. bimAMa- is this supposed to be bimABm?

Line 74. I would recommend saying which gene we are speaking of bimA, bimABp, or bimABm?

Line 151. What is "ace"?

line 186 &211. This is gene, correct? Should be "b".

Reviewer #3: Minor revision

Reviewer #4: Line 24: with antibiotic treatment?

Line 74: precise the version of the gene

Line 223-224: could be interesting to study this with mutants and wildtype infections

Line 238: are the alleles related to virulence genes?

Table 2: problem in the percentage column

Figure 2: legend title ‘Alignment of…’

**Summary and General Comments**

Reviewer #2: Previous concerns addressed. Just minor editorial items for the authors' consideration.

Reviewer #3: The paper is suitable for publication.

Reviewer #4: The study described in this paper is interesting as it presents the situation of melioidosis in a part of Australia and asociated variants/mutations and also virulence and antimicrobial resistance. The study is well supported by the data included even if some missing data about the strains could have been of great importance to trace back the origin of the strain (but it is sometimes difficult, we all know that).

PLOS authors have the option to publish the peer review history of their article (what does this mean?). If published, this will include your full peer review and any attached files.

Reviewer #2: No

Reviewer #3: No

Reviewer #4: No

Figure Files:

Data Requirements:

Reproducibility:

References

---

## [Editor Report · Decision Letter 2]

24 May 2022

Dear Dr. Harris,

We are pleased to inform you that your manuscript 'Clinical Burkholderia pseudomallei isolates from north Queensland carry diverse bimABm genes that are associated with central nervous system disease and are phylogenomically distinct from other Australian strains' has been provisionally accepted for publication in PLOS Neglected Tropical Diseases.

Best regards,

Adly M.M. Abd-Alla, Prof asso.

Associate Editor

Abdallah Samy

Deputy Editor

---

## [Editor Report · Acceptance letter]

8 Jun 2022

Dear Dr. Harris,

We are delighted to inform you that your manuscript, "Clinical Burkholderia pseudomallei isolates from north Queensland carry diverse bimABm genes that are associated with central nervous system disease and are phylogenomically distinct from other Australian strains," has been formally accepted for publication in PLOS Neglected Tropical Diseases.

Best regards,

Shaden Kamhawi

co-Editor-in-Chief

Paul Brindley

co-Editor-in-Chief
